# Implications of Myocardial Bridge on Coronary Atherosclerosis and Survival

**DOI:** 10.3390/diagnostics12040948

**Published:** 2022-04-10

**Authors:** Roxana Oana Darabont, Ionela Simona Vișoiu, Ștefania Lucia Magda, Claudiu Stoicescu, Vlad Damian Vintilă, Cristian Udroiu, Dragoș Vinereanu

**Affiliations:** 1Department of Cardiology and Cardiovascular Surgery, University of Medicine and Pharmacy “Carol Davila”, 37 Dionisie Lupu, 030167 Bucharest, Romania; stefania.magda@umfcd.ro (Ș.L.M.); claudius_md@hotmail.com (C.S.); vladvintila2005@yahoo.com (V.D.V.); vinereanu@gmail.com (D.V.); 2Department of Cardiology, University Emergency Hospital of Bucharest, 169 Splaiul Independenței, 050098 Bucharest, Romania; simona.visoiu@drd.umfcd.ro (I.S.V.); cristiudroiu@yahoo.com (C.U.)

**Keywords:** myocardial bridge, atherosclerosis, survival

## Abstract

Background: In this study, we aimed to describe the impact of MBs on atherosclerosis and survival, in patients with coronary artery disease (CAD). Methods: We retrospectively studied 1920 consecutive patients who underwent conventional coronary angiography for suspected CAD. Atherosclerotic load (AL), defined as the sum of degrees of stenosis, and general atherosclerotic load (GAL), representing the sum of AL, were compared between patients with MB and a control group without MB; patients in these groups were similar in age and sex. We assessed survival at 10 years after the last enrolled patient. Results: Prevalence of MB was 3.96%, predominantly in the mid-segment of left anterior descendent artery (LAD). In the presence of MB, GAL was lower (158.1 ± 93.7 vs. 205.3 ± 117.9, *p* = 0.004) with a lesser AL in the proximal (30.3 ± 39.9 vs. 42.9 ± 41.1, *p* = 0.038) and mid-segments (8.1 ± 20.0 vs. 25.3 ± 35.9, *p* < 0.001) of LAD. Based on a Multinominal Logistic Regression, we found that the presence of MB on LAD (regardless of its location on this artery) is a protective factor against atherosclerotic lesions, decreasing the probability of significant stenosis, especially of those ≥70%, on the entire artery (B −1.539, OR 4660; 95% CI = 1.873–11.595, *p* = 0.001) and on each of its segments as well: proximal LAD (B −1.275, OR 0.280; 95% CI = 0.015–5.073; *p* = 0.038), mid-LAD (B −1.879, OR 6.545; 95% CI = 1.492–28.712; *p* = 0.013) and distal LAD (B −0.900, OR 2.459, 95% CI = 2.459–2.459, *p* = 0.032). However, 10-year survival was similar between groups (76.70% vs. 74.30%, *p* = 0.740). Conclusion: The presence of MB on LAD proved to be a protective factor against atherosclerosis for the entire artery and for each of its segments, but it does not influence long-term survival in patients with CAD.

## 1. Introduction

Myocardial bridge (MB) represents the muscle fibers that abnormally overlie the intramyocardial passage of an epicardial coronary artery, which consequently becomes tunneled in its path beneath them. It is commonly located in the second segment of the left anterior descending artery (LAD) (70% to 90% of cases). It is recognized as the most frequent congenital coronary anomaly [1,2,3,4,5,6,7]. 

About one third of all MBs can exert systolic compression on the adjacent coronary artery and even fewer have clinical expression [8]. Although an uncommon event, myocardial ischemia induced by isolated MBs may become manifest, sometimes through significant clinical forms such as silent ischemia, stable angina [9,10], acute coronary syndromes [11,12,13,14,15,16,17,18,19,20,21], stress cardiomyopathy [22], supraventricular [23] or malignant arrythmias [24,25,26,27,28], exercise-induced atrioventricular conduction block [29] and even sudden cardiac death [30,31,32,33,34], most of these conditions being revealed by case reports or series of cases. The consequences of MBs are still incompletely understood. It is considered that some of their effects are exerted by special hemodynamic conditions in the injured artery, such as the persistence of systolic artery narrowing during diastole [35,36,37,38,39] or vasospasm in the adjacent segments to the bridge through the induction of endothelial dysfunction [40,41,42,43].

The most debated aspect of the side effects of MBs consists in their impact on the atherosclerotic process, which is perceived as a “double-edged sword” [44]. Anatomical, histological, biological, and imagistic evidence sustains that MBs have a dual effect upon atherosclerosis: protection against disease in the tunneled segment [2,45,46,47,48,49,50,51,52,53,54] and increased AL in the pre-bridge portion compared to the segments crossed by the bridge or those distally from it [8,46,52,53,54,55,56,57,58,59,60,61,62,63,64,65,66,67,68,69,70,71]. It remains controversial to date whether and how MBs influence atherosclerotic load in the presence of obstructive coronary heart disease, especially when it comes to the pre-bridged segment [46,52,55,56,67,68,71,72,73]. Moreover, very few research has targeted the impact of MBs on prognosis in patients with associated obstructive coronary disease [55,57,73]. Therefore, we have conducted a study to find out which of the dual effects of MBs prevails on the atherosclerotic load or on the severity of atherosclerotic lesions in the presence of coronary artery disease (CAD) and weather the presence of MBs influences survival in this category of patients.

## 2. Materials and Methods

We retrospectively studied 1920 consecutive patients who underwent coronary angiography between 2004 and 2010 at our institution, for suspected myocardial ischemia: silent myocardial ischemia (SMI), stable angina, unstable angina, non-ST segment elevation myocardial infarction (NSTEMI) and ST segment elevation myocardial infarction (STEMI). In total, 76 patients with myocardial bridging were identified and enrolled in MB group. For this group, we assigned a control group of 109 patients without MB (non-MB group), randomly selected from all patients with similar age and sex distribution. Cases with severe valvular heart disease or hypertrophic cardiomyopathy, as well as all those with normal coronary arteries, were excluded from the study. The study was conducted in accordance with Declaration of Helsinki. The Institutional Review Board certified that collecting and using the data of the included patients complies with European legislation on personal data protection.

We retrieved from the patients’ medical records data on demographic and clinical characteristics, cardiovascular risk factors (family history, known diagnosis of diabetes mellitus, smoking, dyslipidemia or arterial hypertension, lipid fractions), and coronary angiography. The angiographic diagnosis was defined as systolic narrowing of one of the epicardial coronary arteries, produced by muscle compression (the “milking effect”), with a “step down” and “step up” effect, demarcating the affected area. Atherosclerotic load (AL) was defined as the sum of degrees of stenosis induced by atherosclerotic plaques. We evaluated a segmental AL for: left main coronary artery (LM), left anterior descending artery (LAD), first diagonal artery (D1), second diagonal artery (D2), ramus intermedius (RI), left circumflex artery (LCX), first obtuse marginal branch (OM1), second obtuse marginal branch (OM2) and right coronary artery (RCA). LAD was the bridge-carrying artery in all but one of the MB-group patients. Therefore, we considered it relevant to perform a subsequent focused analysis on each segment of the LAD. We also assessed a general atherosclerotic load (GAL) for the whole coronary artery system as the sum of segmental AL. 

The follow-up was carried out at 10 years from the last enrolled patient, using the information platform of the National Health Insurance House, reviewing deaths from all causes and times of death.

### Statistical Analysis

Statistical analysis was performed using SPSS version 20.0 (IBM Corp., Armonk, NY, USA). Continuous variables were reported as mean ± SD and compared for statistical significance with independent-samples *t* test. Categorical variables were expressed as percentages and compared with Chi-square test. We used paired-samples *t* test to compare 2 continuous variables from the same group. 

A Multinominal Logistic Regression based on a Forward Stepwise Method was used having global atherosclerotic severity of the entire LAD (GAS_LAD_CAT) as dependent variable (0–normal artery, 1: <50% stenosis, 2: 50–69% stenosis, 3: 70–89% stenosis and 4: >90% stenosis) and myocardial bridge on LAD (MB_LAD) as factor, while diabetes mellitus, arterial hypertension, smoking, LDL-cholesterol levels, age and gender were used as covariates, at a 95% CI. Similar analysis was also performed for each segment of LAD (proximal, mid, and distal). 

Kaplan–Meier curves were created to assess survival between groups and were compared by the log-rank test. The univariate Cox proportional hazard regression analysis was used to estimate hazard ratios (HRs) and 95% confidence intervals (CIs) for all-cause mortality. Proportional hazards assumptions over time were verified. A *p* value of <0.05 was considered significant. 

## 3. Results

The prevalence of MB in our study population was 3.96%. Among the 76 patients with MB, 98.7% were in LAD (89.5% of them in the middle segment of LAD) and 1.3% in D1. The average length of MB was 12.4 ± 4.1 mm. The atherosclerotic load within the MB was very low: 4.5 ± 16.8% (Table 1).

There was no difference between the two groups in distribution of smoking, dyslipidemia or level of lipid fractions, diabetes mellitus (DM) or maternal/paternal family history (FH) of cardiovascular disease at enrollment. However, the prevalence of hypertension was lower in the MB group (*p* = 0.012) (Table 2).

Clinically, stable angina was seen more frequently in the MB group (*p* = 0.007), but the frequency levels of SMI, unstable angina, and myocardial infarction, both NSTEMI and STEMI, were similar between the two groups (Table 2). 

GAL was significantly lower in the MB group (158.10 ± 93.70, *p* = 0.004), as was AL in the proximal (*p* = 0.038) and especially in the mid-segments of LAD (*p* < 0.001), being similar between the two groups in the distal segment of the LAD (Table 3, Figure 1). There was also a borderline lower AL in the OM1 (*p* = 0.044) (Table 3).

We must notice that, inside both groups, AL was higher in the proximal segments of the LAD compared with the mid-segments (MB group: 30.3 ± 39.9% vs. 8.1 ± 20.0%, *p* < 0.001; non-MB group: 42.9 ± 41.1% vs. 25.3 ± 35.9%, *p* = 0.003) (Figure 2).

Based on Multinominal Logistic Regression, we found that the presence of MB on LAD is a protective factor against the atherosclerotic lesions (the negative value of the B value). In all four regression models, the presence of the MB (regardless of its location on LAD) is decreasing the probability of significant atherosclerotic stenosis, especially of those more than 70% (Table 4, Table 5, Table 6 and Table 7). 

There were no differences in survival over a 10-year follow-up from the last enrolled patient: 76.70% in the MB group vs. 74.30% in the non-MB group (*p* = 0.740) (Figure 3).

Through univariate Cox regression analysis, MB proved not to predict all-cause mortality. The 10-year all-cause mortality was associated only with age (HR 1.060, 95% CI = 1.026–1.094, *p* < 0.001) and GAL (HR 1.003, 95% CI = 1.001–1.005, *p* = 0.023) (Table 8).

## 4. Discussion

With a prevalence of almost 4% and a predominant location in the mid-segment of the LAD, our results on the frequency and topography of MBs are in line with previous reports based on conventional angiography [74,75,76,77]. 

In the present study, we evaluated patients suspected of myocardial ischemia, with atherosclerotic disease confirmed by coronary arteriography, differentiated by the presence or absence of MBs. 

Most of the studies that have addressed the relation of MBs with atherosclerosis have emphasized the protective effect of their presence beneath the bridge [2,45,46,47,48,49,50,51,52,53,54] in comparison with the proximal and distal segments [8,46,53,54,57,58,59,60,61,62,63,64,65,66,69,70,72,73]. Beyond these acknowledged data, we were interested to identify which of the dual effects of MBs prevails on the atherosclerotic load in patients with CAD. For this purpose, we considered it important to have a control group, similar in age and sex with the MB group, and to analyze data not only about the coronary system in general, but precisely about the bridge-carrying artery and its segments. 

We found a lower GAL in patients with MBs, with a decreased AL in the whole LAD.

Furthermore, we showed that LAD, the artery carrying the MB mostly on its mid-segment, had a lower AL in the proximal segment, with a highly decreased AL in the mid-segment, in comparison with the non-MB group. Based on a Multinominal Logistic Regression, we found that the presence of MB on LAD (regardless of its location on this artery) is a protective factor against atherosclerotic lesions, decreasing the probability of significant stenosis, especially of those ≥70%, on the entire artery and on each of its segments as well. These data should be related to published studies that had similar objectives to ours.

We will first refer to studies that have evaluated the impact of MBs on atherosclerotic lesions in LAD. Sun et al. evaluated retrospectively the medical records of 1500 patients who had received coronary angiography. They concluded that the presence of MB is not an additional risk factor for coronary atherosclerosis [56]. Uusitalo et al. have realized a prospective study on 100 patients investigated by coronary computed tomography angiography (CCTA) for intermediate likelihood of CAD and showed that atherosclerotic burden and the presence of vulnerable plaques were comparable between those with and without MBs [46]. Opposite to these data are the results of two studies that have reported a protective effect of MBs on the entire atherosclerotic burden of the coronary arteries. The earliest study of Stolte et al. indicated a decreased rate of atherosclerotic lesions in the presence of MBs on the LAD [52]. Jiang et al., by defining severe obstructive CAD as one requiring PCI or CABG, have described that after adjusting for sex, age, diabetes mellitus, hypertension, and other risk factors, MB still proved a positive role in preventing severe obstructive CAD [71]. 

Another category of studies refers to the analysis of the impact of MB, particularly its impact on the pre-bridged segment. From this perspective, the results are again contradictory. Based on CCTA evaluation, Wang et al. advanced the hypothesis that superficial MB with a depth <2 mm is negatively associated with significant stenosis proximal to the bridge [73], while Nakaura et al., using the same method of investigation, arrived at the assertion that MB in the mid-LAD is an independent risk factor for coronary atherosclerosis in the proximal LAD [67]. Forthcoming basic research reports support the latter category of results. The pre-bridged segment is assumed to be impacted by high intraparietal pressure and turbulent flow [51,64,67] that injures the endothelial layer [60,61] and stimulates the expression of vasoactive or metabolic mediators at this level [62,63]. There are also some morphological characteristics of the MBs that modulate the atherosclerosis development, related to their depth, length, or thickness, as well as the distance from left coronary ostium to the MB [36,45,53,60,61,68,70,73]. However, these factors can only be amplifiers to an already known susceptibility to atherosclerosis of the proximal part of the LAD [78]. Our data prove the same pattern in the distribution of atherosclerotic lesions between the segments of LAD.

In patients with isolated MB, long-term prognosis is generally good, with an 11-year survival of 98% [77]. We found a 10-year survival of 76.7% in patients with MB and CAD, with no significant difference in survival compared to patients without MB. In fact, a mortality rate in patients with CAD and MB that approached 23% is similar with that reported in SYNTAX Extended Survival Study in patients with three-vessel disease and left main coronary artery disease. The 10-year all-cause death was of 22.2% when patients were completely revascularized by PCI and of 24.3% when they were completely revascularized by CABG [79]. Regarding the outcomes in patients with MB and CAD, data from the literature are scarce. Two studies, with a short follow-up period, have reported a worse outcome in the presence of MBs. One of these studies has included consecutive primary revascularized patients with STEMI. The presence of MB in LAD was associated with lower TIMI III flow and higher rates of in-hospital and 6-month mortality [55]. Another series of consecutive patients with DES on LAD evaluated major cardiac events (MACE) as a composite endpoint, including all-cause death, myocardial infarction, target lesion revascularization, and ischemia driven by target vessel revascularization on a three-year follow-up. More MACE developed in MB-group patients, but not all-cause deaths or myocardial infarction [57]. Thus, our study extends the data from the literature to a longer follow-up period, sustaining that MB has no impact on 10-year survival.

Several limitations of our study should be acknowledged. First, the patients were selected from a single-center database, which does not statistically reflect the data of the general population. The study was conducted at a time when the primary revascularization program in STEMI was just being implemented in our hospital, which may explain the relatively high mortality rates in a long-term follow-up. Third, by accessing the database of the National Health Insurance House, we were not able to record major cardiac events; we could only review deaths from all causes and times of death.

## 5. Conclusions

The presence of MB on LAD proved to be a protective factor against atherosclerosis for the entire artery and for each of its segments, especially in its middle and proximal segments. MBs did not influence 10-year survival in patients with obstructive coronary artery disease. To our knowledge, it is the longest follow-up addressed to the impact of MBs on survival in patients with associated coronary artery disease.

## Figures and Tables

**Figure 1 diagnostics-12-00948-f001:**
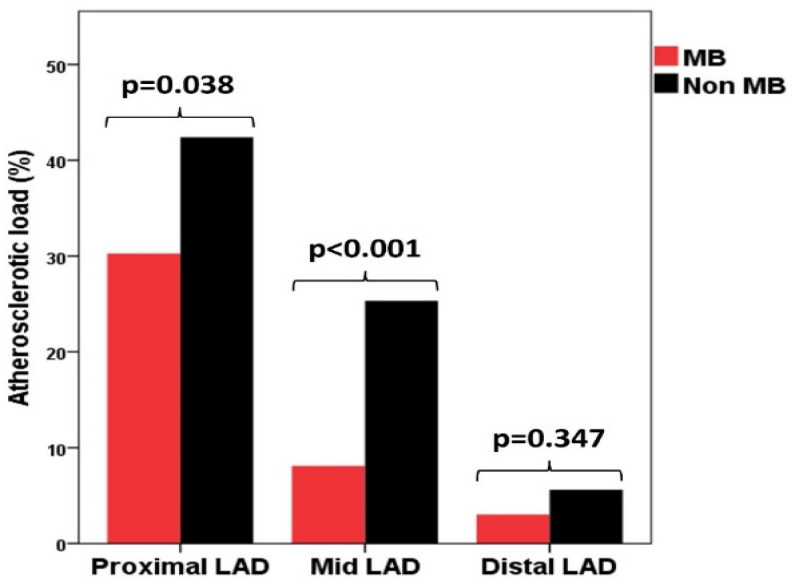
Distribution of atherosclerotic load in the three segments of de left anterior descending artery in MB group and non-MB group. LAD—left anterior descending artery; MB—myocardial bridge.

**Figure 2 diagnostics-12-00948-f002:**
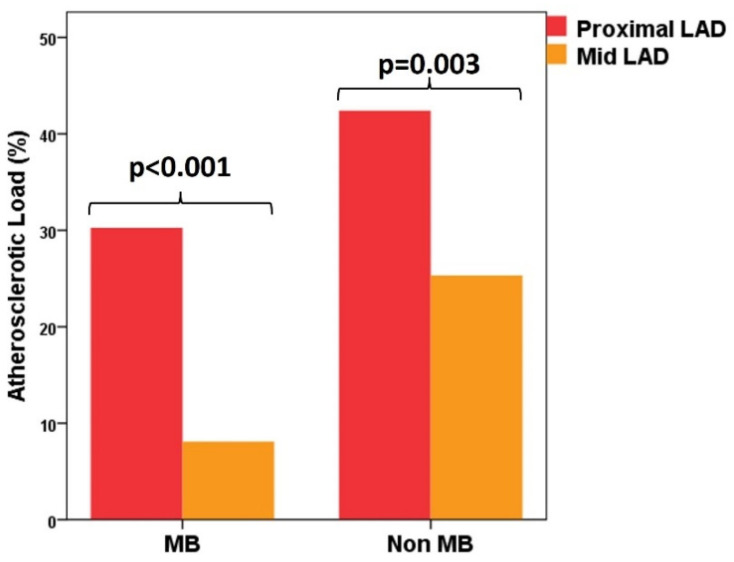
Distribution of atherosclerotic load between the proximal and distal segment of the left anterior descending artery in MB group and non-MB group. LAD—left anterior descending artery; MB—myocardial bridge.

**Figure 3 diagnostics-12-00948-f003:**
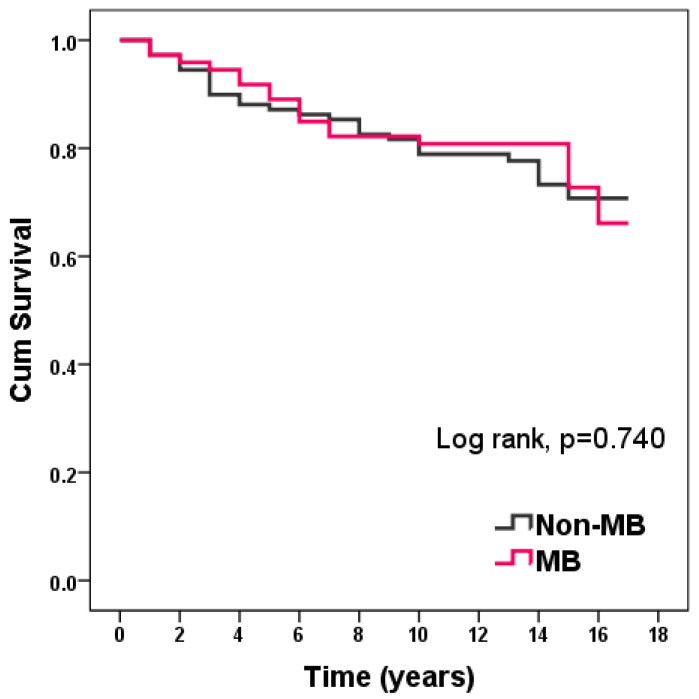
Kaplan–Meier curve of survival in patients with and without MB.

**Table 1 diagnostics-12-00948-t001:** Characteristics of the myocardial bridge.

Characteristics	Values
Prevalence n/total (%)	76/1920 (3.96%)
Location	
LAD (%)	75 (98.7%)
proximal n (%)	3 (3.9%)
mid n (%)	68 (89.5%)
distal n (%)	4 (5.3%)
D1 n (%)	1 (1.3%)
Length (mm)	12.4 ± 4.1
Atherosclerotic load (%)	4.5 ± 16.8

LAD—Left anterior descending artery; D1—first diagonal artery.

**Table 2 diagnostics-12-00948-t002:** Baseline demographic associated cardiovascular risk factors, and clinical characteristics of patients.

Variable	MB (n = 76)	Non-MB (n = 109)	*p* Value
Age (years, mean ± SD)	57.2 ± 10.2	54.5 ± 8.4	0.052
Male (%)	89.5	87.2	0.632
Maternal FH (%)	9.2	3.7	0.117
Paternal FH (%)	11.8	4.6	0.066
Smoking (%)	59.2	67.0	0.280
Hypertension (%)	63.2	79.8	0.012
Dyslipidemia (%)	68.4	66.1	0.821
DM (%)	6.6	20.2	0.315
Hyperlipidemia burden			
LDL (mg/dL)	117.1 ± 43.8	127.3 ± 43.4	0.193
HDL (mg/dL)	37.5 ± 8.9	37.9 ± 9.1	0.791
TG (mg/dL)	168.8 ±118.5	194.7 ± 117.5	0.210
Clinical presentation			
SMI (%)	1.3%	5.5%	0.142
Stable angina (%)	26.3%	11.0%	0.007
Unstable angina (%)	28.9%	43.1%	0.050
NSTEMI (%)	6.6%	7.3%	0.842
STEMI (%)	38.2%	33.0%	0.472

FH—family history; DM—diabetes mellitus; LDL—low-density lipoprotein; HDL—high-density lipoprotein; TG—triglycerides; SMI—silent myocardial ischemia; NSTEMI—non-ST segment elevation myocardial infarction; STEMI—ST segment elevation myocardial infarction.

**Table 3 diagnostics-12-00948-t003:** Atherosclerotic load in patients with and without MB.

Variable	MB	Non-MB	*p* Value
AL-LM	3.8 ± 16.6	6.8 ± 19.2	0.257
AL-LAD	41.4 ± 45.6	73.3 ± 53.8	<0.001
AL-LAD proximal	30.3 ± 39.9	42.9 ± 41.1	0.038
AL-LAD mid	8.1 ± 20.0	25.3 ± 35.9	<0.001
AL-LAD distal	3.0 ± 15.9	5.6 ± 19.7	0.347
AL-D1	11.3 ± 28.1	11.7 ± 28.9	0.920
AL-D2	1.8 ± 12.3	2.4 ± 14.7	0.775
AL-RI	5.5 ± 20.9	2.3 ± 12.9	0.198
AL-LCX	33.5 ± 46.5	36.3 ± 48.0	0.694
AL-OM1	5.7 ± 20.3	14.2 ± 32.4	0.044
AL-OM2	6.5 ± 22.7	3.6 ± 18.1	0.328
AL-RCA	44.1 ± 42.8	55.4 ± 39.7	0.068
GAL	158.1 ± 93.7	205.3 ± 117.9	0.004

AL—atherosclerotic load; LM—left main coronary artery; LAD—left anterior descending artery; D1—first diagonal artery; D2—second diagonal artery; RI—ramus intermedius; LCX—left circumflex artery; OM1—first obtuse marginal branch; OM2—second obtuse marginal branch; RCA—right coronary artery; GAL—general atherosclerotic load.

**Table 4 diagnostics-12-00948-t004:** Regression model for the probability of atherosclerotic stenosis on the entire left anterior descending artery in the presence of myocardial bridges.

GAS_LAD_CAT ^a^		B	SE	*p*	OR	95% Confidence Interval for OR	
						Lower Bound	Upper Bound
<50% STENOSIS	Intercept	−0.172	0.509	0.735			
	MB	0.721	0.667	0.279	0.486	0.132	1.795
50–69% STENOSIS	Intercept	−2.219	0.842	0.008			
	MB	−1.438	0.725	0.047	4.211	1.017	17.425
70–89% STENOSIS	Intercept	−1.784	0.642	0.005			
	MB	−1.539	0.465	0.001	4.660	1.873	11.595
>90% STENOSIS	Intercept	−0.385	0.468	0.411			
	MB	−0.902	0.404	0.025	2.465	1.117	5.437

^a^ The reference category is: NORMAL ARTERY. Model Fitting Information: −2 Log Likelihood: 146.81; Chi-Square: 10.267; *p*: 0.036; Cox and Snell: 0.163; Nagelkerke: 0.173; McFadden: 0.061; B: regression coefficient; SE: standard error; OR: odds ratio; GAS_LAD_CAT: categories of global atherosclerotic severity on entire LAD; MB: myocardial bridge.

**Table 5 diagnostics-12-00948-t005:** Regression model for the probability of atherosclerotic stenosis on the proximal segment of left anterior descending artery in the presence of myocardial bridges.

GAS_LAD proximal_CAT ^a^		B	SE	*p*	OR	95% Confidence Interval for OR	
						Lower Bound	Upper Bound
<50% STENOSIS	Intercept	−17.960	1.105	0.000			
	MB	−0.785	1.096	0.474	0.456	0.053	3.905
50–69% STENOSIS	Intercept	−17.172	0.777	0.000			
	MB	−0.255	0.828	0.058	0.775	0.153	3.926
70–89% STENOSIS	Intercept	−0.486	1.467	0.741			
	MB	−1.275	1.479	0.038	0.280	0.015	5.073
>90% STENOSIS	Intercept	0.035	1.402	0.980			
	MB	−1.064	1.432	0.045	0.345	0.021	5.711

^a^ The reference category is: NORMAL ARTERY. Model Fitting Information: −2 Log Likelihood: 95.048; Chi-Square: 11.201; *p*: 0.008; Cox and Snell: 0.673; Nagelkerke: 0.723; McFadden: 0.345; B: regression coefficient; SE: standard error; OR: odds ratio; GAS_LAD proximal_CAT: categories of global atherosclerotic severity on proximal LAD; MB: myocardial bridge.

**Table 6 diagnostics-12-00948-t006:** Regression model for the probability of atherosclerotic stenosis on the mid-segment of left anterior descending artery in the presence of myocardial bridges.

GAS_LAD mid_CAT ^a^		B	Std. Error	*p*	OR	95% Confidence Interval for OR	
						Lower Bound	Upper Bound
<50% STENOSIS	Intercept	−2.079	0.395	0.000			
	MB	−0.473	0.575	0.411	0.623	0.202	1.923
50–69% STENOSIS	Intercept	−3.332	0.709	0.000			
	MB	−0.934	0.809	0.248	2.545	0.522	12.418
70–89% STENOSIS	Intercept	−3.332	0.709	0.000			
	MB	−1.879	0.754	0.013	6.545	1.492	28.712
>90% STENOSIS	Intercept	−4.025	0.994	0.000			
	MB	−1.761	1.059	0.046	5.818	0.730	46.378

^a^ The reference category is: NORMAL ARTERY. Model Fitting Information: −2 Log Likelihood: 85.027; Chi-Square: 36.458; *p*: 0.002; Cox and Snell: 0.180; Nagelkerke: 0.211; McFadden: 0.103; B: regression coefficient; SE: standard error; OR: odds ratio; GAS_LAD mid_CAT: categories of global atherosclerotic severity on mid-LAD; MB: myocardial bridge.

**Table 7 diagnostics-12-00948-t007:** Regression model for the probability of atherosclerotic stenosis on the distal segment of left anterior descending artery in the presence of myocardial bridges.

GAS_LAD distal_CAT ^a^		B	Std. Error	*p*	OR	95% Confidence Interval for OR	
						Lower Bound	Upper Bound
<50% STENOSIS	Intercept	−17.647	0.634	0.000			
	MB	−0.845	0.725	0.244	0.430	0.104	1.781
50–69% STENOSIS	Intercept	−19.694	2.016	0.992			
	MB	−0.900	0.000	0.235	2.459	2.459	2.459
70–89% STENOSIS	Intercept	−19.694	2.016	0.992			
	MB	−0.900	0.000	0.032	2.459	2.459	2.459
>90% STENOSIS	Intercept	−1.043	1.337	0.435			
	MB	−2.527	1.065	0.048	0.080	0.010	0.645

^a^ The reference category is: NORMAL ARTERY. Model Fitting Information: −2 Log Likelihood: 35.129; Chi-Square: 11.095; *p*: 0.035; Cox and Snell: 0.590; Nagelkerke: 0.120; McFadden: 0.093; B: regression coefficient; SE: standard error; OR: odds ratio; GAS_LAD distal_CAT: categories of global atherosclerotic severity on distal LAD; MB: myocardial bridge.

**Table 8 diagnostics-12-00948-t008:** Univariate Cox regression analysis of variables associated with 10-year all-cause mortality.

Variable	HR (95% CI)	*p* Value
Myocardial bridge	0.904 (0.495–1.652)	0.743
Age	1.060 (1.026–1.094)	<0.001
Male	1.117 (0.473–2.640)	0.800
Smoking	0.867 (0.477–1.574)	0.639
Hypertension	1.886 (0.877–4.057)	0.105
Dyslipidemia	2.642 (0.359–19.443)	0.340
DM	1.915 (0.938–3.910)	0.075
LDLc	0.993 (0.984–1.002)	0.143
GAL	1.003 (1.001–1.005)	0.023
AL-LAD	1.001 (0.995–1.006)	0.752

DM—diabetes mellitus; AL—atherosclerotic load; GAL—general atherosclerotic load.

## Data Availability

Upon request from the corresponding author.

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
