# Peer review of "Implications of Myocardial Bridge on Coronary Atherosclerosis and Survival"

_diagnostics, 2022, doi:10.3390/diagnostics12040948_

Round 1

Reviewer 1 Report

Comments to the Author

The Authors present the study design entitled “Implications of myocardial bridge on coronary atherosclerosis and survival”.

The topic is very interesting, could have an important clinical impact, and be helpful for doctors in their everyday clinical practice.

The Authors aimed to compare two groups of patients: with MB and age- and the sex-matched control group of patients without MB in coronarography. The sample size is large, seems to be efficient to perform the appropriate analyzes and to formulate the appropriate conclusions. The introduction part is well written, the methodology is clear. The reviewer appreciates the Authors’ effort and notices the important clinical value of the study.

Otherwise, the reviewer has some comments which need to be risen before my final decision:

Major comments:

  1. The precise explanation of AL and GAL needs to be added. The Authors have to explain exactly how they calculate those parameters. “We defined AL as the average percentage of narrowing induced by atherosclerotic plaques” – what does it mean exactly, it could be not clear for the average reader. The additional table with AL, segmental AL, and GAL in the whole group, MB+ and MB- group could be necessary. If the AL is the average percentage of narrowing, and GAL is the average of AL, why GAL is above 100%? Is GAL a sum of segmental AL? It is not clear.
  2. The statistics are not efficient. It is necessary to perform univariate (possibly multivariate) Cox regression (proportional hazards regression) analyses for investigating the effect of MB (contrary to others presented in the table 2 parameters).
  3. The discussion part should be re-written. The Authors should perform that part according to their results despite reviewing the literature per se. The Authors should contrast their results with the data from the literature and try to explain the possible differences (or similarities).
  4. It is not clear what is the real novelty of the study contrary to almost one hundred other studies.Minor comments:1.      The conclusions should be concentrated on the precise conclusions, without redundant text. 2.      In some places there is a lack of citation: after the second sentence in the Introduction part (“It is located commonly…”) and in the last paragraph of the Introduction (… few studies have addressed the impact of this anomaly on the severity of atherosclerotic burden and prognosis…” – please cite that few studies). 

Reviewer 2 Report

This interesting paper studies the prevalence and impact of myocardial bridging on clinical outcome. 

The study is well designed and results are well presented but need serious revision and improvement of the quality of writing. 

1) the introduction section is too long and needs to be better focused.

2) Likewise the discussion is full of chats that need to be made concise and straight to the points in order not to lose the readers concentration.

3) There are many types that need careful revision e.g.

MB location is more de 5 cm from ostium 

4) the second paragraph in th conclusion section is redundant and should be deleted.

5) Authors should better address the issue of MB protection vs promotion of proximal LAD atherosclerosis with better reasoning 

Round 2

Reviewer 2 Report

The revised draft of the paper has significantly improved its quality, however there are still some careless English and typo mistakes that need careful attention

Author Response

Thank you very much for recommendations! We hope that we have identified and corrected them. The manuscript was language proofed by a third part expert.